# Healthy Diets from Sustainable Food Systems: Calculating the WISH Scores for Women in Rural East Africa

**DOI:** 10.3390/nu15122699

**Published:** 2023-06-09

**Authors:** Gudrun B. Keding, Jacob Sarfo, Elke Pawelzik

**Affiliations:** Division Quality of Plant Products, Department of Crop Sciences, University of Goettingen, 37073 Göttingen, Germany; jsarfo@gwdg.de (J.S.); epawelz@gwdg.de (E.P.)

**Keywords:** sustainable diets, healthy diets, food systems, WISH, East Africa, protective food groups, limiting food groups, dietary diversity

## Abstract

Diets should be healthy for the benefits of both humans and the environment. The World Index for Sustainability and Health (WISH) was developed to assess both diets’ healthiness and environmental sustainability, and the index was applied in this study. Food intake quantities for single foods were calculated based on the data collected from four 24-h recalls during two seasons in 2019/2020 with women of reproductive age in two rural areas each in Kenya, Tanzania and Uganda (*n* = 1152). Single foods were grouped into 13 food groups, and the amount of each food group consumed was converted to an overall WISH score and four sub-scores. The food groups with a low WISH score were fruits, vegetables, dairy foods, fish, unsaturated oils and nuts, meaning that their consumption was outside the recommended range for a healthy and sustainable diet. Contrariwise, the intake of red meat and poultry was partly above the recommended intake for those women who consumed them. The overall WISH score and sub-scores showed that the consumption of “protective” food groups needed to increase in the study population, while the consumption of “limiting” food groups was sufficient or should decrease. For future application, we recommend dividing food groups that are critical for nutrition, e.g., vegetables, into sub-groups to further understand their contribution to this index.

## 1. Introduction

Our current food systems are unsustainable to a large extent and are both drivers of and affected by human diets, malnutrition and diet-related non-communicable diseases [1]. Often, only one perspective is taken, and diets are seen as either a driver of change in food systems, such as through changing consumer demand, or as an outcome of food systems, for example through climate change constraints or changing food environments [2]. Similarly, when studying food systems, only environmental [3,4] or nutritional aspects [5,6,7] are typically considered, while there is a gap in combining both to understand and assess nutritional and environmental trade-offs and the synergies of different food systems. Environmental impact of food production was measured in several studies, and reviews about sustainable diets have been available since 2014 [8]. A new index, the World Index for Sustainability and Health (WISH), was recently developed to assess diets in terms of both environmental sustainability and healthiness [9]. Furthermore, two other indices have been proposed to explicitly assess adherence to the EAT-Lancet planetary health diet [10]. One of these indices uses binary scoring criteria [11], while the other one uses gradual scoring similar to the WISH; yet, their reference values are not in grams, but caloric density is applied, i.e., the energetic contribution of 2500 kcal/d based on the EAT-Lancet reference diet is used [12]. While these two indices only take the EAT-Lancet planetary health diet as a basis, the WISH rates the impact of different food groups on the environment based on a study by Clark et al. (2019) [13] and the recommended intake in g/d of the Global Burden of Disease Study [14]. Especially because it considers the latter study, the WISH, which has been applied to only a few datasets so far, was chosen to be applied to our dataset on the food consumption of women of reproductive age in Kenya, Tanzania and Uganda during two different seasons. Women were chosen as they are a group particularly affected by malnutrition. The data were collected in the framework of a larger project on “Fruits and Vegetables for All Seasons” (FruVaSe), which aimed to improve rural development and nutrition in the three countries through promoting resource-efficient processing techniques for surplus fruits and vegetables to bridge seasonal gaps. The project selected three fruit and three vegetable species, which had been underutilized in research and development, to develop different products from them. Therefore, a key issue was the contribution of processed fruits and vegetables to nutrition security and how nutritional values, tastes and presentational characteristics can be retained with the shelf-life prolonged [15,16,17,18,19]. Product development was accompanied by nutritional assessment to understand the contribution these fruit and vegetable products could make to the standard diet [20]. Next to nutrient adequacy, the environmental sustainability of the standard diet was also of interest.

## 2. Materials and Methods

Data collection was carried out during two seasons in 2019 and 2020 in Kenya (Kitui and Taita Taveta counties), Tanzania (Morogoro and Mtwara regions) and Uganda (Jinja and Kayunga districts). The research areas were purposely selected according to the key fruits and vegetables grown in these areas, while households for the survey were randomly selected based on village and household lists, as described in a previous study [21]. Four non-consecutive 24-h recalls, with two recalls during each season, were conducted with women of reproductive age, together with an individual survey on socio-economic information. Surveying started with 600 households per country during the first season, but not all women were available for the second interview; after data cleaning, 445 women in Kenya, 292 in Tanzania, and 415 in Uganda were included in the data analysis, for a total of 1152 women.

Food intake quantities for single food items were calculated based on the participants’ specifications and, in the case of mixed dishes, based on Kenyan food recipes [22] and the Tanzanian food composition table [23]. For further details, please see Sarfo et al. (2021) [21]. All single food items were then grouped into the 13 food groups used for calculating the WISH (Table 1). In the next step, the amount of each food group consumed was converted to an overall WISH score by giving a score between 0 and 10 based on both the healthiness of the food group consumed and environmental sustainability [9]. Accordingly, each participant received a component score for each food group (0–10) and a total WISH score (0–130). In addition, the authors of the WISH suggested calculating four sub-scores since a diluting effect is anticipated for the total WISH score. These sub-scores were also calculated for our data (see Table 1 for the classification of “healthiness” and “impact on environment”), including a Healthy sub-score (0–100; summing 8 protective and 2 neutral food groups); a Less healthy sub-score (0–30; summing 3 limiting food groups); a Low-environmental-impact sub-score (0–60; summing 6 low-environmental-impact food groups); and a High-environmental-impact sub-score (0–70; summing 4 medium- and 3 high-environmental-impact food groups) [9].

**Table 1 nutrients-15-02699-t001:** Component consumption when compared to the recommended intake and WISH scoring for women in Kenya, Tanzania, Uganda and East Africa as a whole (pooled).

Dietary Component (Healthiness/Impact on Environment)	Non-Consumers (%)				Intakes of Food Groups for All Participants in g Mean (SD)				Recommended Intake in g/d (Lower and Upper Range of Intake) ^1^	Direction of Change in Intake to Obtain Higher WISH Score ^2^
	Kenya (N = 445)	Tanzania (N = 292)	Uganda (N = 415)	East Africa (Pooled Data; N = 1152)	Kenya (N = 445)	Tanzania (N = 292)	Uganda (N = 415)	East Africa (Pooled Data; N = 1152)		
Whole grains (protective/low)	0.2	1.0	6.3	2.6	90.0 (49.9)	86.3 (40.6)	84.7 (58.3)	87.1 (51.0)	≥125 (100–150)	Increase
Vegetables (protective/low)	0	0	0	0	120.7 (71.6)	126.3 (70.5)	161.1 (111.5)	136.6 (89.7)	300 (200–600)	Increase
Fruits (protective/low)	75.7	63.0	31.1	56.4	13.3 (33.6)	34.0 (73.0)	51.2 (95.2)	32.2 (72.9)	200 (100–300)	Increase
Dairy foods (protective/medium)	0.7	95.5	46.7	41.3	67.5 (39.9)	4.1 (20.9)	75.7 (97.1)	54.4 (70.6)	250 (0–500)	Increase
Red meat (limit/high)	69.4	72.9	68.7	70.1	17.2 (42.4)	15.0 (37.3)	11.8 (30.8)	14.7 (37.3)	14 (0–28)	Adequate
Fish (protective/high)	94.8	19.2	37.8	55.1	0.5 (2.8)	30.2 (30.6)	10.4 (21.8)	11.6 (23.4)	28 (0–100)	Increase
Eggs (neutral/medium)	97.8	100.0	94.9	97.3	0.2 (3.6)	0.0 (0.0)	0.6 (4.9)	0.3 (3.7)	13 (0–25)	Increase
Chicken/other poultry (neutral/medium)	98.9	96.6	96.1	97.3	0.3 (3.1)	1.5 (8.8)	1.3 (7.3)	1.0 (6.5)	29 (0–58)	Increase
Legumes (protective/low)	7.4	31.8	27.7	20.9	89.3 (65.2)	58.7 (75.4)	38.6 (55.3)	63.3 (68.3)	75 (0–100)	Increase
Nuts (protective/medium)	100	80.8	15.7	64.8	0 (0)	3.1 (10.2)	25.0 (35.9)	9.8 (24.9)	50 (0–75)	Increase
Unsaturated oils (protective/low)	0	0	1.9	0.7	34.1 (12.8)	19.6 (43.8)	8.1 (7.2)	21.1 (26.3)	40 (20–80)	Adequate/Increase
Saturated oils (limit/high)	100	55.5	89.9	85.1	0 (0)	2.5 (4.3)	0.4 (1.6)	0.8 (2.6)	11.8 (0–11.8)	Adequate
Added sugars (limit/low)	0	3.1	1.0	1.1	16.8 (10.8)	13.0 (6.0)	44.1 (20.0)	25.7 (19.8)	31 (0–31)	Adequate/Decrease

Yellow background color refers to food groups that are consumed by nearly all participants. ^1^ Recommended intake according to the Global Burden of Disease Study [14] and as suggested by [9]. ^2^ Red color refers to a change from Table 2 (consumers only).

**Table 2 nutrients-15-02699-t002:** Component consumption by consumers only when compared to the recommended intake and WISH scoring for women in Kenya, Tanzania, Uganda and East Africa as a whole (pooled data).

Dietary Component	Intakes of Food Groups for Participants Who Consumed during the Last 24 h in g, Mean (SD)								Recommended Intake in g/d (Lower and Upper Range of Intake) ^1^	Direction of Change in Intake to Obtain Higher WISH Score ^2^
	Kenya (Total N = 445)		Tanzania (Total N = 292)		Uganda (Total N = 415)		East Africa (Pooled Data) (Total N = 1152)			
	N		N		N		N			
Whole grains	444	90.2 (49.8)	289	87.2 (39.8)	389	90.4 (55.8)	1122	89.5 (49.7)	≥125 (100–150)	Increase
Vegetables	445	120.7 (71.6)	292	126.3 (70.5)	415	161.1 (111.5)	1152	136.6 (89.7)	300 (200–600)	Increase
Fruits	108	54.8 (48.8)	108	92.0 (95.4)	286	74.3 (106.9)	502	73.9 (95.4)	200 (100–300)	Increase
Dairy foods	442	68.0 (39.6)	13	91.2 (44.7)	221	142.1 (90.8)	676	92.7 (70.4)	250 (0–500)	Increase
Red meat	136	61.0 (56.1)	79	55.6 (53.9)	130	37.8 (45.3)	345	49.1 (54.5)	14 (0–28)	Decrease
Fish	23	9.3 (8.7)	236	37.3 (29.8)	258	16.7 (25.7)	517	25.8 (29.2)	28 (0–100)	Increase
Eggs	10	10.0 (22.7)	0	0	21	12.7 (18.5)	31	11.8 (19.6)	13 (0–25)	Increase
Chicken and other poultry	5	29.0 (0)	10	44.8 (18.3)	16	33.7 (17.9)	31	36.5 (17.2)	29 (0–58)	Adequate/Decrease
Legumes	412	96.5 (62.4)	199	86.1 (77.3)	300	53.4 (58.7)	911	80.0 (67.5)	75 (0–100)	Adequate/Increase
Nuts	0	0	56	16.2 (18.3)	350	29.7 (37.3)	406	27.8 (35.6)	50 (0–75)	Increase
Unsaturated oils	445	34.1 (12.8)	292	19.6 (43.8)	407	8.2 (7.2)	1144	21.2 (26.4)	40 (20–80)	Adequate/Increase
Saturated oils	0	0	130	5.7 (4.9)	42	4.3 (3.2)	172	5.4 (4.6)	11.8 (0–11.8)	Adequate
Added sugars	445	16.8 (10.8)	283	13.4 (5.6)	411	44.5 (19.6)	1139	26.0 (19.7)	31 (0–31)	Adequate/Decrease

Yellow background color refers to food groups that are consumed by nearly all participants. ^1^ Recommended intake according to the Global Burden of Disease Study [14] and as suggested by [9]. ^2^ Red color refers to a change from Table 1.

Moreover, we looked into the diversity of fruits and vegetables consumed and grouped them according to their nutrient content, such as vitamin A. The grouping was performed based on the guidelines for creating a dietary diversity score for women (FAO 2021), namely into (i) dark-green leafy vegetables, (ii) other vitamin A-rich fruits and vegetables, (iii) other vegetables, and (iv) other fruits. In addition, the ratio of dark-green leafy vegetables to total vegetables consumed was calculated.

As the overarching FruVaSe project focused on fruit and vegetable processing, the consumption of processed foods and their relevance—both positive and negative—for a healthy and sustainable diet was of interest. The WISH only includes “whole grains”, but, in our study population, many foods made from refined grains were consumed. Therefore, we compared the intake of whole grains to refined grain products, as well as to the intake of tubers and starchy vegetables, which are also not considered in the WISH.

## 3. Results

### 3.1. Food Group Intake

The food groups from which foods were consumed by nearly all women are whole grains, vegetables, unsaturated oils and added sugars (Table 1). For the other food groups, only a certain share of women consumed these foods during the last 24 h (mean intake in two seasons, i.e., four days) and, therefore, the mean intakes were also calculated among “consumers” only (Table 2). An increase in consumption to obtain a higher WISH score is suggested for all but four food groups. These are red meat and saturated oils, which consumption is within the suggested range; unsaturated oils, which consumption is within the range but could increase especially for women in Tanzania and Uganda; and added sugars, which consumption is also within the range, but it should decrease especially for women in Uganda. When considering consumers only, whereby women who consumed a certain food group during the study period were taken into account, the intake of red meat is clearly above the recommended intake and should decrease, the intake of chicken is above the recommended mean intake for women in Tanzania and Uganda, while the intake of legumes is adequate, although it should increase for women in Uganda (Table 2).

### 3.2. The WISH Score and Sub-Scores

The 13 food groups from which the WISH scores are calculated are shown in Figure 1. Here, the mean score obtained by rural women in East Africa for each food group is depicted. A higher score (maximum of 10) shows a more healthy and sustainable consumption of the respective food group. The mean score reaches 10 (or nearly 10 in one case) for eggs, chicken/other poultry and saturated oils for all three countries and East Africa (pooled data). It also reaches 10 for added sugars for Tanzania and 9 for added sugars for Kenya, while it reaches about 8 for red meat for all three countries and for East Africa as a whole.

The food groups with a low WISH score are vegetables, fruits, dairy foods, fish, nuts and unsaturated oils. This means that the consumption of these food groups is not within the suggested range and is much lower for the women in our study, so that this “under-consumption” contributes to an unhealthy and unsustainable food consumption in general. Whole grains and legumes have mixed results across the three countries (Figure 1).

Next to the total WISH score, four sub-scores were calculated. The mean scores for each country and for East Africa as a whole are shown against the maximum score possible in Figure 2. The higher these scores are, the higher the compliance with the recommendations for the respective foods (protective, limiting, low or high environmental impact). Consequently, the higher the sub-scores are, the healthier the diets (for the two health sub-scores) and the lower the impact on the environment (for the two environmental impact sub-scores) [9]. While the mean Less healthy sub-score is close to its maximum for all countries, the other three scores are far from the maximum score, especially the Low-environmental-impact sub-score in Uganda and the Healthy sub-score in all three countries (Figure 2).

As food consumption data during two different seasons were collected, the scores were also compared between these two seasons. The “on-season” was the season when a key fruit or vegetable was available in abundance, whereas the “off-season” meant that a particular fruit or vegetable was hardly available. With this focus on key fruits and vegetables, seasonality did not necessarily apply to other foods, such as staple foods or animal source foods. From Figure 3 and Figure 4, it can be seen that there are significant differences between the two seasons for East Africa (pooled data) for the total WISH scores and sub-scores, namely for the WISH scores of all components, except those of fish, eggs, poultry, legumes and unsaturated oils, as well as the High-environmental-impact sub-score. However, the differences are not very pronounced, and this is also true for the data from each of the three countries. Therefore, in the following section, only data from both seasons (mean values) are shown and discussed.

The authors of the WISH suggested, i.a., that it could be useful to divide red meat into ruminant (such as beef) and non-ruminant (such as pork) categories [9]. We tested this for our data; however, in the non-ruminant category, only two women were listed from one county in Kenya who consumed 16 g/d of pork each, while all other “red meat” consumption was beef (consumed in all study areas and during both seasons) and goat, which were consumed in Uganda and in one area each in Kenya and Tanzania, and offal, which was consumed in Uganda and in one area in Kenya. Offal products were not specified regarding which animals they were derived from, but most products are commonly from cow.

When the intake of fruits and vegetables was further divided into four groups, the women in our study consumed the largest amount from the group “other vegetables”, which includes onion, tomato, cabbage and eggplant, among others. To some extent, “dark green leafy vegetables” were consumed, although in very little amount in Uganda, namely at a ratio of 10% when compared to total vegetable consumption. This was different from Kenya (42%) and Tanzania (52%), and the share of dark-green leafy vegetables for all three countries together (pooled data) was 22%. “Other vitamin A rich fruits and vegetables”, namely carrot, mango, passion fruit, pawpaw and pumpkin, were hardly consumed during both seasons (Figure 5).

### 3.3. Other Foods and Food Groups Not Captured by the WISH 

The WISH does not include any foods from refined grains, which, compared to whole grains, are finely ground, have bran and germ removed and have a comparably low mineral content. Additionally, roots, tubers and starchy vegetables are not considered in the WISH. These are, however, important food groups in the rural communities of this study, and therefore, we listed all foods within these groups to compare them with the group of whole grains, which is included in the WISH (Table 3). In comparison to the three grains which are commonly used as whole grains, i.e., maize, millet and sorghum, in different dishes, the list of foods from refined grains is much longer and mainly includes foods and snacks which are fried in oil. Additionally, the consumption of roots and tubers and snacks made from these foods is high for the whole study population, although some of these foods are mainly found in one country, such as cocoyam is found only in Uganda, and cassava and yam are found only in Uganda and Kenya. Participants who did not consume any roots or tubers were in fact only a few women in Uganda (2%), while in Kenya and Tanzania, only a few participants did not consume any refined grains (11% and 5%, respectively) (Table 4). The consumption of foods from the refined grain group and the root and tuber group is clearly higher than for whole grains when looking at the pooled data, with some differences between the countries (Table 4). The consumption of roots and tubers of nearly 600 g/d is particularly high in Uganda, while the consumption of snacks made from refined grains is rather high in Kenya (nearly 130 g/d) and Tanzania (nearly 190 g/d).

The recommended intake is according to Table 1 for whole grains, according to the EAT-Lancet Commission 2019 for roots/tubers, and according to the Kenya National Guidelines for Healthy Diets 2017 for refined grains (clear suggestion only for pregnant women; there is a trade-off between the limitation of refined grains and the fortification of the same foods: “In Kenya, extra vitamins and minerals are added into commercially produced maize meal and wheat flour through fortification. Fortified starchy foods help people to get more vitamins and minerals into their diet.” [24]).

## 4. Discussion

Checking diets both for their healthiness and their environmental friendliness has been approached by several projects and groups [10,11,12,13]; however, the process is complex and a holistic view still missing in many policy and project interventions. In this study, we applied the WISH, one of the newly established indices to assess diets regarding their healthiness and environmental friendliness, to dietary intake data from rural women in East Africa. We chose the WISH because it is not solely based on data from the EAT-Lancet planetary health diet [10] but also takes into account data from the Global Burden of Disease study [14] for the suggested range of g/d for each food group. Basing the calculations on kcal intake as performed by another index [12] allows the assessment of different calorie scenarios, which is not possible with the WISH. However, the WISH gives more credit to micronutrients and other bioactive compounds, which are better depicted through the g/d intake of certain food groups, e.g., fruits and vegetables.

### 4.1. Food Intake of Single Food Groups

Before we looked at the WISH scores, we compared the intake of single food groups to the recommendations and determined whether the intake should increase or decrease in order to obtain a higher WISH score. If the whole study population (pooled data for East Africa) was taken into account, the intake of all food groups needs to increase, except for food groups that should be limited for healthiness, namely red meat, saturated oils and added sugars, which were consumed in sufficient amounts by our study population. Interestingly, if we only analyzed the data from women who consumed particular foods, red meat and chicken (the latter being classified as “neutral” in terms of healthiness) should decrease in consumption while the intake of legumes should increase. These results are in line with studies on nutrition transition in Sub-Sahara Africa, which has even reached some women in rural areas nowadays and is evidenced through an increased consumption of sugars, oils and fats, animal source products and, in general, highly or even ultra-processed food products [25,26,27,28,29]. Oil consumption, both of saturated and unsaturated oils, was not too high and could even increase for healthy and protective unsaturated oils in our study population. However, the amount consumed was underestimated in this calculation as food products based on refined grains, which were often fried in oil, were not considered, although their consumption was much higher than for whole grains, in particular in Kenya and Tanzania. This is a limitation as these fried products have been associated with higher BMI values in the same study population and, in fact, with overweight and even obesity among women [21]. It seems that to calculate a more meaningful WISH score, in particular regarding the healthiness level, refined grains and their products, as well as starchy roots and tubers, need to be integrated into the score as a separate dietary component. 

While it is, in general, suggested to limit the consumption of refined grains, there is a trade-off; in Kenya, for example, maize and wheat flour is fortified with several vitamins and minerals [24], which would increase nutrient intake. However, fortified flours mainly reach consumers in urban areas, while food fortification in rural areas is rather ineffective because of a weak industrial infrastructure [30]. In areas where mainly one’s own or locally produced cereals without fortification are consumed, as it is the case in our study areas, the suggestion to eat more whole grains and limit the consumption of refined grains still hold true. Nevertheless, depending on the population group that is surveyed, the WISH would also need to give credit to the fortification of food products.

Similarly, processed foods and their relevance for and impact on healthy and sustainable diets are not considered in detail by the WISH and the underlying data were not collected by the EAT-Lancet Commission; for example, vegetables, fruits and legumes included in the score can be fresh but also frozen, cooked, canned or dried [9]. Processed foods are often seen in association with increased levels of saturated fats, trans-fats, free sugars and salt/sodium, which is especially true for ultra-processed foods, and their consumption should be limited. At the same time, it has been suggested that less incentives should be provided to the food industry for the production of processed foods with high levels of the above-mentioned components [31], or even that regulations are needed. On the other hand, there is little discussion on how minimally processed foods, e.g., through drying, canning or fermentation without the addition of many other ingredients, could contribute to bridge seasonal gaps, in particular for highly perishable foods, increase food safety and contribute to lowering the waste of surplus foods during season.

### 4.2. The WISH and Sub-Score Results: Healthiness and Environmental Sustainability of East African Diets

The WISH scores reached the highest score of 10, meaning optimal consumption, for the food groups of eggs, chicken/poultry and saturated oils for the pooled data (all three countries together). While the consumption of these food groups can remain the same, there is a challenge for those food groups with an extremely low WISH score, such as fruits (0.8), nuts (1.3), vegetables (1.7) and fish (2.5). While there is no upper limit for fruits and vegetables and, consequently, a low score means a low amount consumed, both a lower and an upper limit of consumption were set for nuts and fish: as both are considered protective foods, a consumption of 0 g/d did not score any points, which is similar to the consumption for nuts above the amount of 75 g/d and for fish above 100 g/d, as an increased pressure on the environment through a higher intake is assumed [9]. Thus, a low score can mean both under- or overconsumption of these food groups, which is also true for the food groups of dairy foods, legumes and unsaturated oils. On the contrary, whole grains, fruits and vegetables have no upper limit in consumption, while red meat, eggs, chicken/poultry, saturated oils and added sugars have no lower limit, and their scores are, thus, easier to interpret. When going back to our data, it, however, becomes clear that, for our study population, too little consumption of all food groups with both lower and upper limits is a challenge and an increase to reach a sustainable and healthy diet is badly needed. A low consumption of fruits and vegetables is found in many other studies [25,29,32,33,34], and their consumption, which is critical for sustainable food and nutrition security cannot be overemphasized, while fruit and vegetable breeding, production, processing and consumption still has not obtained sufficient support in research and development [35,36].

The average total WISH score of 62 is nearly half of the maximum total score and much higher than the score of 46 for the Vietnamese population in the original study by Trijsburg et al. (2021) [9], which, however, included both women and men, and not only women as in our case. The Kenyan women in the study areas performed slightly better in terms of healthy and sustainable food intake, with a WISH score of 65, when compared to the women in Tanzania (63) and Uganda (59). Nevertheless, as it has been mentioned above, some food groups with a high WISH score can counterbalance others with a low score, so that additionally calculating the sub-scores is a better approach. From these four sub-scores, only the Less healthy sub-score was close to its maximum (25 out of 30) for the overall study population, being highest for the Tanzanian and Kenyan women (27 each) and lowest for Uganda (21), meaning that the three limiting food groups of red meat, saturated oils and added sugars were consumed more or less within the suggested recommendations. This is in line with the consumption data for single food groups when all study participants were taken into account; however, this was not the case when only consumers were considered as some food groups, in particular red meat, were then consumed above the recommended intake. We report here the data from four non-consecutive 24-hour recalls collected during two different seasons; yet, this data collection period could still lead to some bias when, for example, red meat was consumed only on one particular day and otherwise not during the rest of the week. Still, meat consumption in general has risen in Sub-Sahara Africa and the daily average per capita consumption is already above the suggested 70 g/d by the EAT-Lancet Commission [29]. At the same time, the amounts of plant-based foods and, in particular, fruits and vegetables remain under the suggested recommendations by the World Health Organization of 400 g per capita per day for Sub-Sahara Africa as a whole [29] and, likewise, for our study population in East Africa, which also depicted a very low Healthy sub-score of 38 on average (out of 100).

This corresponds with a small Low-environmental-impact sub-score of only 19 on average (out of 60), when the consumption of whole grains, legumes, unsaturated oils and added sugars is summarized next to the consumption of fruits and vegetables, because of the low consumption of these food groups in our study population in general. Additionally, the High-environmental-impact sub-score reached only 43 on average (out of 70), although some of the food groups included in this score have upper limits and a high consumption would also result in a low sub-score. Again, it is important to check the individual WISH score of each food group when interpreting the overall score. At the same time, the combination of food groups in one score helps to compare food consumption among population groups or even countries. In our study, when compared to the other two countries, Uganda has a much smaller Low-environmental-impact sub-score of only 12 (out of 60) and the largest High-environmental-impact sub-score of 47 (out of 70), which should be taken into account in future food system activities. Suggestions for transformations of food systems and for increasing both dietary and environmental health (to maximize all sub-scores) should include reducing the consumption of animal source food products while increasing the amounts of plant-based foods in diets [37]. In particular, proteins from the expected increase in meat consumption in sub-Saharan Africa (SSA) should be substituted with plant-based alternatives or insect-based proteins, which are either already accepted or have shown growing interest in SSA [38]. While animal-source food consumption will and can persist worldwide, food production needs to shift from intensive and fossil fuel-based systems toward diverse and mixed crop–livestock systems or, if appropriate, even aquaculture or livestock–agroforestry systems [37,39].

### 4.3. The WISH and Food-Based Dietary Guidelines

Both a healthy and balanced diet and environmental effects of food consumption are considered in some food-based dietary guidelines (FBDGs), which exist currently for 98 countries in the world. In Africa, so far, only nine countries have developed and published FBDGs [40], such as Kenya [24], while they are in preparation for Tanzania [41] and under discussion in Uganda [42].

The FBDG for Kenya encourages consumption of some foods rated as “protective” in terms of healthiness and “low impact on environment”, such as guidelines to eat five servings of fruits and vegetables per day and to consume legumes, nuts and seeds at least four times a week. It is also recommended to “include whole or unprocessed starchy foods” in general; yet, in the key messages, “whole grains” are not mentioned, but it is suggested to “choose fortified maize meal and wheat flour”. Additionally, for the “protective” and “low environmental impact” food group related to unsaturated oils, no clear recommendation is given, only that “oils are healthier than fats” and that oils and fats should be used in moderation. For saturated oils, it is clearly stated to “limit consumption”; this is the same for the WISH, which, as mentioned above, derives its classification from the EAT-Lancet recommendations [10] and the Global Burden of Disease Study [14]. The guidelines are also in line with the WISH recommendation for added sugars, which have a low impact on the environment but should be limited in consumption and are suggested to be “used sparingly”. The Kenyan guidelines are less detailed regarding animal-source foods as they combine red meat, fish, poultry and eggs and recommend consumption twice a week with no upper limit. In addition, insects as a source of protein are mentioned, which are not included in the WISH so far. Dairy food, or more precisely milk, fermented milk or yoghurt, is recommended to be consumed every day, which, according to the WISH, should also have an upper limit.

### 4.4. The WISH and Dietary Diversity within Food Groups

The variation of foods within one food group and, therefore, nutrient availability is high and influences nutrient intake [43]. Therefore, although an evaluation at the food group level is needed for simplicity and to obtain a quick overview, variety within food groups needs to be taken into account, in particular in extremely diverse groups such as fruits and vegetables.

Clear recommendations for the consumption of different fruit and vegetable types—other than to eat varied types in general—are not yet available. However, it is acknowledged that different types have different nutritional benefits, for example, dark-green leafy and cruciferous vegetables as well as citrus and dark-colored fruits have superior effects on the outcomes of chronic diseases compared to others (Wallace et al., 2020) [44]. Based on our data, we observed that the largest amount consumed was for “other vegetables”, which was dominated by onion, tomato, cabbage and eggplant. Dark-green leafy vegetables, which are known for their high beta-carotene content and have been associated with a reduction in the risk for type II diabetes [45], were consumed in much smaller amounts of only about 50 g/d on average. While other studies have calculated the ratio of dark-green or red/orange vegetables to total vegetables [12], an extra group for “dark green vegetables” and for “red and orange vegetables” should be created. Another suggestion made by Trijsburg et al. (2021) [9] would be to group “green leafy vegetables”, “vitamin A rich fruits/vegetables” and “citrus fruits”. Because of the very different compositions of fruits vs. vegetables, e.g., higher amount of sugar in fruits, and their various processing methods before consumption, e.g., of fruit juices, and, consequently, their different health effects [46,47], we suggest separating the fruit and vegetable groups and highlighting in particular “dark green vegetables” as one stand-alone food group. 

This is similar to the meat group, where it is also differentiated between red meat and chicken/other poultry because of the different health and environmental impacts. Vegetables should be consumed to a much larger extent and, together with fruits, should make up about half of the food amount, while meat is suggested to be either not consumed at all or only have a small share [9,10,12]. Therefore, a shift to differentiate more between various vegetable and fruit types in terms of contribution to a healthy diet is badly needed. As differentiation will be different on a local basis since different types of fruits and vegetables are available locally, sorting according to colors is a good basis for a global application and has already been suggested as a rapid assessment of the food environment. The so-called Produce Color Diversity Tool [48] records the type and number of fresh fruits and vegetables according to their color category. This can, in turn, be used to identify the availability of secondary metabolites or dietary phytochemicals, which give colors to particular plant products. Most dietary phytochemicals have health claims, such as antioxidant for beta-carotene, lutein and anthocyanin [49,50,51,52]; protection against carcinogens for lycopene [53]; and cancer chemo-protective effects for betalain [52]. While a higher color diversity in diet is associated with a healthier diet, the differentiation of fruits and vegetables according to color should be taken up in future nutritional assessment and nutritional education tools.

While the differences in terms of health impacts are clear, the differences in terms of environmental impacts for different fruit and vegetable types are not yet considered in the calculations so far [13]. While differences in environmental impact are obvious when comparing, for example, ruminant meat with pork production, with ruminant meat production having higher agricultural inputs per unit of meat produced and having higher methane emission [54], differences between fruit and vegetable cropping systems are less clear. The same fruit or vegetable can be produced with higher or lower environmental impact; however, when compared to other foods, a general assumption of low environmental impact is made, regardless of the fruit or vegetable type [13]. This is applicable to our study population as fruit and vegetable production in the study areas is small-scale, mostly rain-fed production with a low input, while intensive large-scale vegetable cropping and fruit orchards with a high input of fertilizers and pesticides would need to be classified in a different “impact on environment” category.

## 5. Conclusions

The WISH and its sub-scores show that in the investigated areas in rural East Africa, the consumption of “protective” food groups, such as fruits, nuts and vegetables, need to increase, while the consumption of “limiting” food groups is sufficient or should even decrease. Due to a high number of non-consumers of some food groups, in particular animal-source foods, the picture is partly biased. The WISH does not include any foods from refined grains and no roots, tubers or starchy vegetables as well. These are, however, important food groups in the rural communities of this study and would need to be considered in the score to show a complete picture. Similarly, it is suggested to divide complex and critical food groups, such as vegetables, into further sub-groups to understand their contribution to this index. Dietary diversity as a key to both healthy and sustainable diets should be considered not only among food groups but also within each food group, which could be attained by considering the different colors of foods during dietary assessment, in particular the colors of fruits and vegetables.

In general, the WISH and its sub-scores allow the differentiation between the overall healthiness and environmental sustainability of diets in one country and comparison between countries. Local food-based dietary guidelines could even “learn” from the WISH and adapt their recommendations accordingly to not only include healthiness of foods but also environmental impact. Therefore, checking single food groups for consumption details will always be necessary and, for future consumption and food system studies, there is no way around analyzing single food and food group consumption and raising awareness about the importance of dietary diversity as a basis of sustainable diets. Still, the overall WISH score and sub-scores will be crucial assessment tools as the current environmental and nutritional conditions call for a transformation of food systems, which requires increasing attention to and application of environmental and dietary health scores. At large, a transformation of the food system in East African countries requires the local governments in charge, both in the health and agriculture sector, to place greater emphasis on reducing or at least not increasing the consumption of animal-source foods, while increasing the intake of plant-based foods for both human and environmental health.

## Figures and Tables

**Figure 1 nutrients-15-02699-f001:**
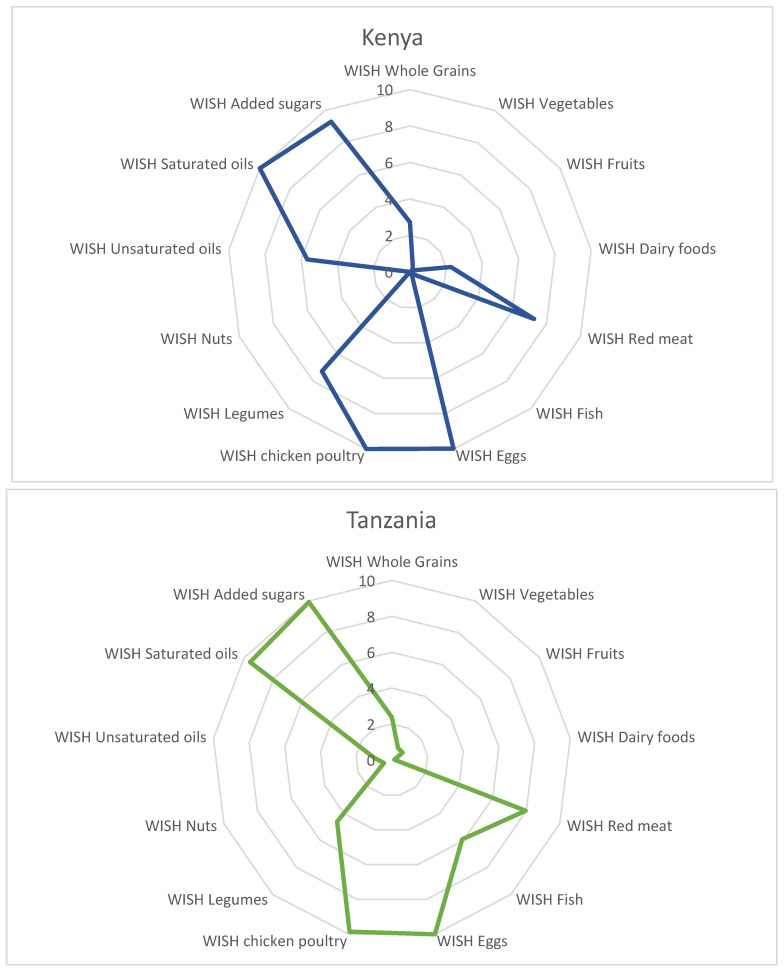
Mean scores of the components of the WISH for rural women in Kenya (N = 445), Tanzania (N = 292), Uganda (N = 415) and East Africa as a whole (pooled data, N = 1152).

**Figure 2 nutrients-15-02699-f002:**
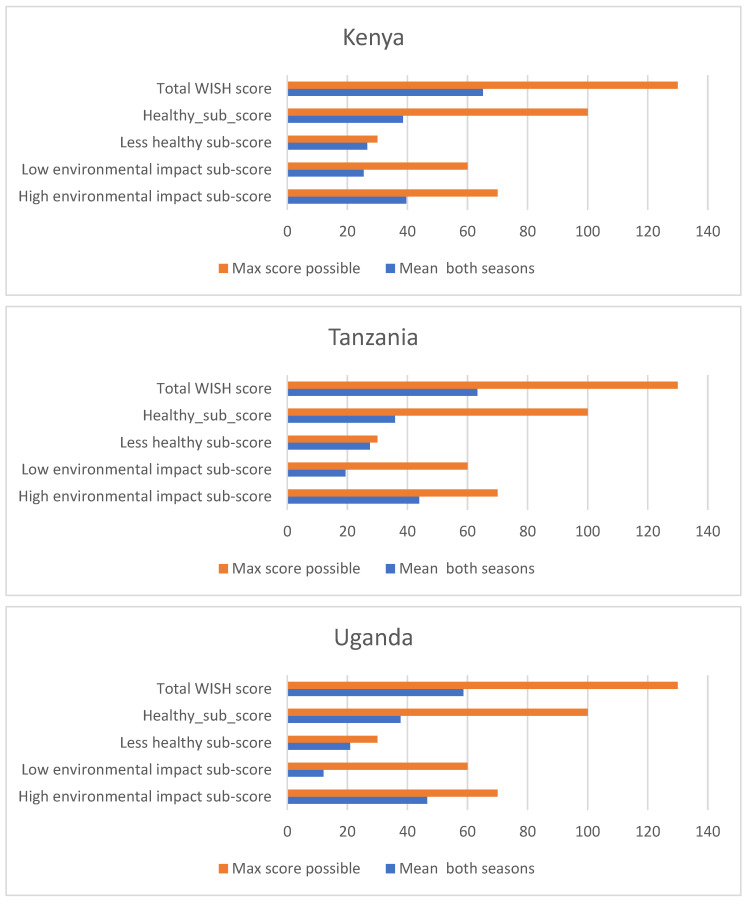
Total WISH scores and sub-scores for women in Kenya (N = 445), Tanzania (N = 292), Uganda (N = 415) and East Africa as a whole (pooled data, N = 1152), compared to the maximum score possible.

**Figure 3 nutrients-15-02699-f003:**
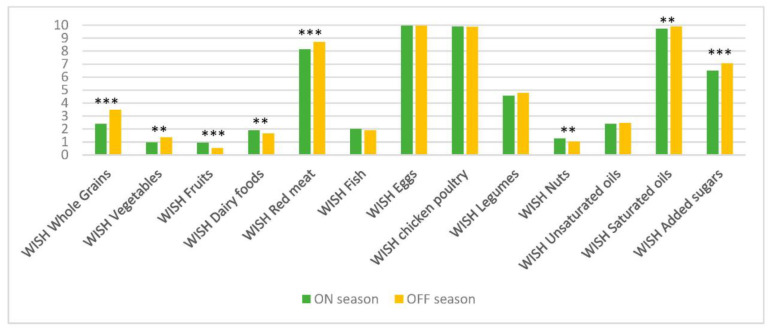
Mean scores of the components of the WISH for rural women in East Africa (pooled data, N = 1152) during two different seasons. ** and *** represent statistical significance at *p* < 0.01 and *p* < 0.001, respectively.

**Figure 4 nutrients-15-02699-f004:**
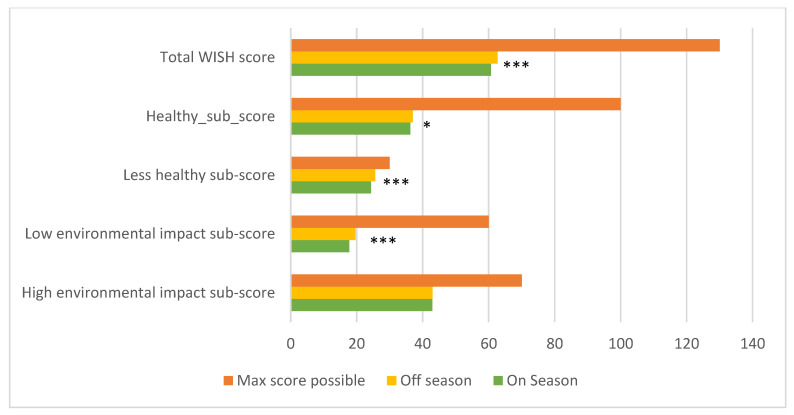
Total WISH scores and sub-scores for women in East Africa (pooled data) during two different seasons and with both seasons combined, compared to the maximum score possible. *, and *** represent statistical significance at *p* < 0.05, and *p* < 0.001, respectively (between off-season and on-season).

**Figure 5 nutrients-15-02699-f005:**
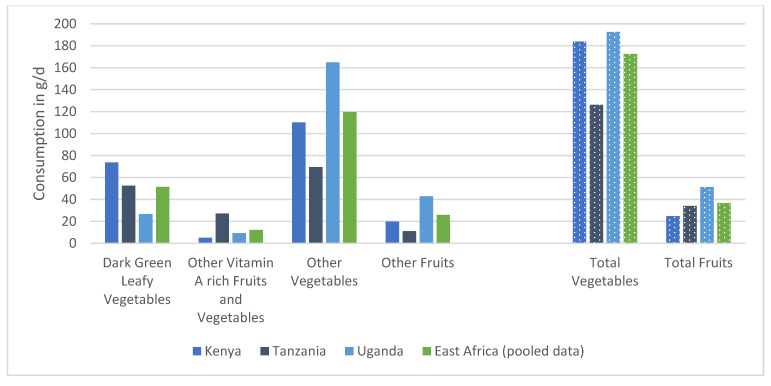
Consumption in g/d (mean of four non-consecutive days) of different vegetable and fruit groups by women from rural Kenya (N = 445), rural Tanzania (N = 292), rural Uganda (N = 415) and all three countries together as pooled data (N = 1152).

**Table 3 nutrients-15-02699-t003:** Foods and food products within different starchy food groups as reported by women in Kenya, Tanzania and Uganda.

Captured by the WISH	Not Captured by the WISH	
Whole Grains	Refined Grains or Snacks Made from the Same Foods Plus Other Ingredients	Roots/Tubers and Snacks Made from the Same Foods
Maize (flour) Millet (flour) Sorghum (flour)	4. Bread/buns 5. Cake 6. Cornflakes 7. Hardcorn (maize and oil) 8. Noodles 9. Pilau masala (rice with spices) 10. Popcorn 11. Rice (flour) 12. Samosa (pastry filled with vegetable or meat, fried in oil) 13. Wheat flour products fried in oil (chapatti, halfcake, mandazi, pancake and vitumbua) 14. Wheat flour	15. Bagiya (cassava and soyabean flour, fried in oil) 16. Cassava (flour) (*Manihot esculenta*) 17. Cocoyam/taro (*Colocasia esculenta*) 18. Irish potato (*Solanum tuberosum*) 19. Potato chips 20. Cooking banana/matoke (*Musa* spp.) 21. Sweet potato (*Ipomoea batatas*) 22. Yam (*Dioscorea* spp.)

Exact ingredients of these refined grain products are not known, which is a limitation as these foods and their ingredients, e.g., oils and fats, were left out in the overall food amount calculation based on the WISH.

**Table 4 nutrients-15-02699-t004:** Consumption of selected starchy components compared to the recommended intake in Kenya, Tanzania, Uganda and East Africa as a whole (pooled data).

Dietary Component	Non-Consumers (%)				Intakes of Food Groups for All Participants in g, Mean (SD)				Recommended Intake in g/d (Lower and Upper Range of Intake) ^1^
	Kenya (N = 445)	Tanzania (N = 292)	Uganda (N = 415)	East Africa (Pooled Data; N = 1152	Kenya (N = 445)	Tanzania (N = 292)	Uganda (N = 415)	East Africa (Pooled Data; N = 1152)	
Whole grains	0.2	1.0	6.3	2.6	119.5 (56.1)	86.3 (40.6)	84.7 (58.3)	98.5 (55.9)	≥125 (100–150)
Refined grains (products)	11.2	4.8	14.9	10.9	128.0 (159.7)	185.6 (838.8)	49.4 (46.1)	114.3 (437.4)	limit
Roots/tubers	59.6	33.9	2.2	32.4	46.2 (97.6)	81.9 (115.9)	588.6 (440.4)	250.6 (376.0)	50 (0–100)

^1^ Recommended intake according to the Global Burden of Disease Study [14] and as suggested by [9].

## Data Availability

The data presented in this study are available from the corresponding author upon request.

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
