# Peer review of "Healthy Diets from Sustainable Food Systems: Calculating the WISH Scores for Women in Rural East Africa"

_nutrients, 2023, doi:10.3390/nu15122699_

Round 1

Reviewer 1 Report

Thank you for the opportunity to review this manuscript. Specific suggestions for the manuscript are outlined below.

Abstract

-          Please clarify what the purpose/objective of this research was, and what the rationale was for conducting this study.

-          Please indicate the actual timeline (at least years) that this study was conducted.

-          Why were only women included in the study? Please provide more context and rationale in the abstract.

Introduction

-          Please describe the “Fruits and Vegetables for All Seasons” (FruVaSe) project in greater detail and include citations.

Results

-          It would help to use a more objective term to replace “good” in the text and tables presented for the WISH score results (e.g., replace with “adequate” or “in alignment with recommended intakes”).

The manuscript was largely well-written. I suggest making changes to use more objective language, as previously suggested for the results section. 

Author Response

Thank you very much for the review and your very helpful feedback. We tried to address all suggestions and have explained what we changed in the point-to-point reply below.

Reviewer 1

Comment/ suggestions

Reply to reviewer

Abstract

-          Please clarify what the purpose/objective of this research was, and what the rationale was for conducting this study.

Unfortunately only 200 words are allowed for the abstract, so we struggled to add your suggestions. We tried to rephrase the first and second sentence to make our objective more clear, yet are now exceeding the 200 word limit by one word.

-          Please indicate the actual timeline (at least years) that this study was conducted.

The years in which the project took place have been added in line 11.

-          Why were only women included in the study? Please provide more context and rationale in the abstract.

Again, we reached the limit of 200 words already and had to compromise on information. We added “of reproductive age” in line 11 in the abstract and in addition we added more information (“Women were chosen, as they are a group particularly affected by malnutrition.”)  in line 72 in the Introduction.

Introduction

-          Please describe the “Fruits and Vegetables for All Seasons” (FruVaSe) project in greater detail and include citations.

This has been added at the end of chapter 1. Introduction, in lines 76-83 including citations of relevant references.

Results

-          It would help to use a more objective term to replace “good” in the text and tables presented for the WISH score results (e.g., replace with “adequate” or “in alignment with recommended intakes”).

Thank you for this very good suggestion, we replaced “good” in relation to the WISH index everywhere with “adequate”

Comments on the Quality of English Language

The manuscript was largely well-written. I suggest making changes to use more objective language, as previously suggested for the results section. 

The text was revised as suggested above.

Reviewer 2 Report

The paper is well prepared on a solid scientific approach. A few minor edits are noted for consideration by the authors. The conclusion is less strong than it could be based on the profound insights revealed in the paper. A notation to this effect is included in the paper. See attached.

Line 46, p. 4 extend should be extent

Line 131, p. 7 own should be one’s own

Line 162, p. 8 On contrast should be In contrast or On the contrary

Line 196, p. 8, small low should be with a low (remove small)

Line 269, p 10 make off should be make up about

Only a few recommendations made on the use of terms to help convey the meaning more clearly.

It is a well written paper.

Author Response

Thank you very much for the review and your very helpful feedback. We tried to address all suggestions and have explained what we changed in the point-to-point reply below.

Reviewer 2

Comment/ suggestions

Reply to reviewer

The paper is well prepared on a solid scientific approach. A few minor edits are noted for consideration by the authors. The conclusion is less strong than it could be based on the profound insights revealed in the paper. A notation to this effect is included in the paper. See attached.

Thank you, we changed the last paragraph of the conclusions according to your suggestions.

Line 46, p. 4 extend should be extent

Changed accordingly

Line 131, p. 7 own should be one’s own

Changed accordingly

Line 162, p. 8 On contrast should be In contrast or On the contrary

Changed to “on the contrary”

Line 196, p. 8, small low should be with a low (remove small)

“Low environmental impact sub-score” is a fixed term and we want to say here that this score was small – but we also see that it is confusing and therefore we capitalised the “Low” (and also the other sub-scores) to make this more clear.

Line 269, p 10 make off should be make up about

Changed accordingly

Comments on the Quality of English Language

Only a few recommendations made on the use of terms to help convey the meaning more clearly. It is a well written paper.

Thank you, also for the comments directly in the pdf, we made all changes as suggested, also changed the last paragraph of the conclusions, and adapted the text as explained above.

Reviewer 3 Report

The concept of food sustainability has been the subject of research for several decades, and it's a topic that has increasingly occupied the public consciousness in recent years. 

Food sustainability means producing food in a way that protects the environment, makes efficient use of natural resources, ensures that farmers can support themselves, and enhances the quality of life in communities that produce food, including the animals as well as the people. This idea is the driving force behind a movement to address the fact that significantly more resources go into our global food system than come out of it. In this article the authors used the World Index for Sustainability and Health (WISH), Food intake quantities for single food items were calculated based on data from four 24-hour 10 recalls during two seasons with women in two rural areas each of Kenya, Tanzania and Uganda (n=1152). All single food items were grouped into 13 food groups and the amount of each group consumed was converted to the overall WISH index and four sub-scores. Food groups with a low WISH score were fruits, vegetables, dairy foods, fish, unsaturated oils and nuts, meaning that their consumption was not within the recommended range for a healthy and sustainable diet. Contrariwise, the intake of red meat and poultry was partly above the recommended intake for those women  who consumed it. The WISH index and sub-scores showed that the consumption of “protective”  food groups need to increase in the studied population, while the consumption of “limiting” food  groups was sufficient or should even decrease. For future application, the authors  recommend to divide diverse food groups critical for nutrition, such as vegetables, into sub-groups to understand their contribution to this index. The paper is interesting.

Author Response

Thank you very much for this summary and your feedback.